# Social Distance Monitoring Approach Using Wearable Smart Tags

**Tareq Alhmiedat** [1,2,*] and **Majed Aborokbah** [3,*]

1 Department of Information Technology, Faculty of Computers & Information Technology, University of Tabuk, Tabuk 71491, Saudi Arabia
2 Industrial Innovation & Robotics Center, University of Tabuk, Tabuk 71491, Saudi Arabia
3 Department of Computer Science, Faculty of Computers & Information Technology, University of Tabuk, Tabuk 71491, Saudi Arabia
* Correspondence: t.alhmiedat@ut.edu.sa (T.A.); m.aborokbah@ut.edu.sa (M.A.)

**Abstract:** Coronavirus has affected millions of people worldwide, with the rate of infected people still increasing. The virus is transmitted between people through direct, indirect, or close contact with infected people. To help prevent the social transmission of COVID-19, this paper presents a new smart social distance system that allows individuals to keep social distances between others in indoor and outdoor environments, avoiding exposure to COVID-19 and slowing its spread locally and across the country. The proposed smart monitoring system consists of a new smart wearable prototype of a compact and low-cost electronic device, based on human detection and proximity distance functions, to estimate the social distance between people and issue a notification when the social distance is less than a predefined threshold value. The developed social system has been validated through several experiments, and achieved a high acceptance rate (96.1%) and low localization error (<6 m).

**Keywords:** social distance; localization; monitoring; COVID-19

## 1. Introduction

COVID-19 belongs to a big family of viruses that normally causes moderate to mild upper-respiratory tract ailments. It was first reported in Wuhan, China, at the end of December 2020. The World Health Organization (WHO) has declared COVID-19 as a pandemic, and a global coordinated effort is required to stop the spread of the virus. The transmission of COVID-19 remains unclear, though evidence from other viruses indicates that the disease may spread through direct or indirect contact with an infected person.

During the ongoing COVID-19 disaster, the Internet of Thing (IoT) has played a significant role in a diverse range of healthcare applications. In general, IoT networks consist of a number of small-size, low-cost, and low-power consumption devices that can be attached to any person or be embedded in any object.

Social distancing is critical for people who are at a higher risk for severe illness from COVID-19. Social distancing is the maintenance of a safe distance of at least 1 m from other people in indoor and outdoor spaces to minimize the spread of the virus. It also limits close contact with others in outdoor and indoor spaces, as people can spread the virus before they know that they are sick [1].

Recently, social distancing was proven to be an effective practice to minimize the spreading of COVID-19. Therefore, social distancing has prompted researchers and developers to find technological solutions in order to fight against the spread of the COVID-19 virus. Several mobile applications and IoT devices have been developed recently to work against the spread of COVID-19.

Due to the nature of the virus and the high spread rate, either indoor or outdoor, when human contact exceeds the predefined social distance space, this work presents a system that will assure and monitor the social distance between individuals during runtime with

an accuracy of 98% using a smart localization system. The proposed system has been evaluated through several experimental studies. This experiment followed a number of steps, including the technological aspect of building the system (hardware and software), starting with face detection techniques, then gathering and sending information to access points to alert for crowing in the specific area, which include a number of functions to evaluate the spaces and identify whether this obstacle is a person or something else.

In addition, the proposed system was experimentally studied to test the functionality and usability of the proposed device. Usability studies take into account user acceptance, user comfortability, and device operation. The results showed a high acceptance rate of (96%) and a high ease of use rate of (93.3%), whereas the functional and hardware operation of the device were at a very acceptable level 94% of the time. The main contributions of this paper lie in the following aspects:

A. Reviewing the existing smart social distance monitoring systems developed recently to prevent the spread of COVID-19.
B. Developing a social distance system that allows the user to monitor social distances between people.
C. Testing the efficiency of the developed tags through employment in a large commercial mall in the city of Tabuk.
D. Evaluating the developed social distance system through studying the users' acceptability, overall performance, localization accuracy, and power consumption.

The subsequent sections are organized as follows: Section 2 categorizes and discusses the recently developed social distance systems. In Section 3, the system design is presented and discussed, whereas Section 4 shows the experimental results after conducting several experiments to assess the performance of the developed system. Finally, Section 5 draws a conclusion and future research.

## 2. Related Works

Many digital tools are being explored and developed to contain the spread of COVID-19. In this section, we discuss the existing social distance monitoring and alerting systems. The existing solutions can be categorized into two categories, as presented (Figure 1): wearable social distance systems and standalone social monitoring systems. The former requires attaching a tag to a person (user) to estimate the distance to the surrounding people. In contrast, the latter is based on stationary or mobile devices employed to monitor the social distances between people in the area of interest, based on image analysis methods.

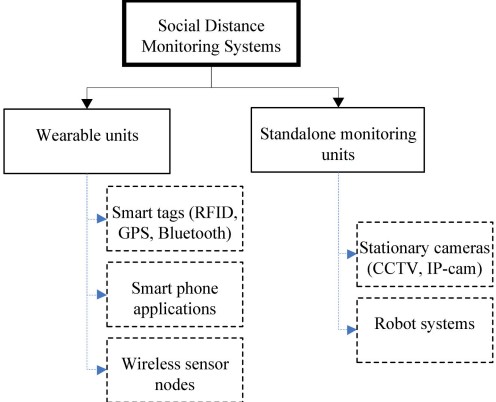

**Figure 1.** Categorization of social distance monitoring systems.

First, the wearable-based systems are considered based on one of the following approaches: smart tags (RFID, GPS, or Bluetooth), smartphone applications (IOS or Android), and wireless sensor nodes. These methods need to be attached to a user to perform distance measurements and emit warning notifications when any user is located in a crowded place.

Cunha et al. developed a prototype of a compact and low-cost wearable electronic device, which is based on the received signal strength (RSS) of the Wi-Fi signals emitted by other wearable devices of the same type, and then estimates the proximity distance between the users and issues a notification when the distance between the users is less than a predefined threshold value [2]. The work presented by Bian and colleagues involves designing a wearable, oscillating magnetic field-based proximity sensing system to monitor the social distances between people. The authors developed a small-size node that was able to detect the distance of the persons approaching in the range of (1.5–2.0 m), where the developed nodes were evaluated in controlled lab and in real-life large store area [3]. The proposed system achieves a detection range beyond 2 m and is robust enough for everyday environments.

Kobayashi et al. proposed a social distance monitoring system for students on a university campus to prevent the spread of COVID-19 infections [4]. The proposed system consists of ESP32-based microcontroller nodes distributed among students in order to allow access to the campus. The distances between the students were calculated by periodically transmitting and receiving Bluetooth Low Energy (BLE) advertising packets between the nodes. The nodes' locations were estimated using signals from the university Wi-Fi network.

Neelavathy et al. proposed smart social distance (SSD) mobile application-based monitoring that can predict the social distances between two persons, assisted by mobile Bluetooth and a mobile camera [5]. The SSD application includes two main steps to predict the social distance. First, it identifies the pedestrians in the video frames using deep learning, and second, it estimates the distance between the two pedestrians through image processing techniques. The developed application is also able to measure the distance using BLE by calculating the received signal strength.

The work presented by Munir and colleagues includes a risk-sensitive social distance recommendation system to guarantee private safety from the COVID-19 pandemic [6]. The authors employed a Bluetooth beacon for a personal area network (PAN), ensuring a reliable distance measurement from other users nearby. The authors formulated a risk-sensitive social distance recommendation problem through incorporating conditional value at risk, which can efficiently discretize the risk of a distance movement for a person, where the formulated problem is solved in a linear model, and they proposed a two phase algorithm.

Rajasekar proposed an IoT-based automated tracking system to identify possible contacts using cost-effective RFID tags and a mobile phone as an RFID reader [7]. The proposed system enables the tracking of people with COVID-19 using a mobile application run on the background and an RFID tag attached to the person infected by COVID-19. As soon as the RFID tag crosses a mobile phone, it is recorded, and the collected details from the mobile phone are passed to the edge device for further processing. The authors employed high frequency passive RFID tags with a range of 1 m, and these tags were classified as passive and active based on the availability of the power source.

An IoT-based social distance monitoring system proposed by Jahmunah and colleagues comprises a mobile phone application and a wearable device [8]. The mobile application consists of a set of contact tracking applications that can establish data collection and interpretation. Lubis proposes a proximity-based COVID-19 contact tracing system using BLE technology, which can trace and control the spread of COVID-19 in the local community [9]. The developed device records the proximity between people, and then data is synchronized using their smartphone.

Alrashidi proposed a system of placement and relocation of people in an indoor environment using an intelligent method based on two optimizers (ant colony and particle swarm) to determine the optimal relocation of a set of students equipped with IoT tags to control their locations and movements [10]. The author employed mini IoT ESP32 M5StickC nodes in their experiment testbed, and completed several experiments.

On the other hand, several wireless sensor network-based localization and tracking systems have been developed to track mobile targets indoors [11–13], which can be employed to estimate the distance between people indoors, and hence warns users in crowded areas. The Received signal strength indicator (RSSI) function can be adopted to measure the distance between the sensor nodes, therefore locating the presence of crowded areas.

Secondly, standalone social monitoring systems have been considered based on stationary or mobile devices distributed in the area of interest to measure the social distances between people through analyzing the images from fixed or mobile digital cameras. This includes digital cameras and robot systems. Ahmed et al. proposed a deep learning platform for social distance tracking using the YOLO v3 object recognition paradigm to recognize humans in video sequences and to estimate the distance between people in open-space areas [14].

The work presented by Ahmed and colleagues includes a social distance monitoring framework based on deep learning architecture as a precautionary step, which assists in maintaining, monitoring, managing, and reducing the physical interaction between an individual in a real-time top view environment [15]. Rahim et al. proposed an efficient solution for real-time social distance monitoring in low light environments [16]. The YOLO v4 algorithm was employed and trained on the ExDARK dataset. A ToF camera was used to observe people at fixed camera distance and to show the resultant distance in real-world units. The proposed solution was able to offer warnings for safety distance violations. The experimental results showed that the YOLO v4 algorithm offered the best detection results in low-light environments, with a 97.84% mAP score.

A smart monitoring physical distances system was proposed by Al-Khazraji, which can monitor people with respect to their physical distances, and offers them suitable feedback [17]. The proposed system detects the number of people available in a specific area and estimates their distances. Then, the system displays warning messages to alert the person who is not respecting the social distancing.

Bashir and colleagues proposed a low-cost Internet of Things-enabled system that counts the number of people entering and leaving a vicinity to ensure physical distances and monitor body temperatures [18]. The system consists of multiple sensor nodes communicating with a centralized server. The authors employed the MobileNet-SSD detection system using the Caffe model in OpenCV. For hardware implementation, the authors adopted Raspberry Pi as a socket server, and the ESP8266 module as a sensor node.

The work presented by Yang and colleagues includes an artificial intelligence system-based real-time social distancing detection and warning system [19]. The proposed system has been validated across real-world datasets to measure the system's generality and performance. The authors employed a fixed monocular camera to detect individuals in a region of interest (ROI) and to measure the interpersonal distances in real-time.

On the other hand, robot platforms have been employed to educate and warn people in common areas about the importance of maintaining social distancing. An autonomous surveillance robot platform was developed to promote social distancing between people [20], composed of social distance detection, urban navigation, and intelligent voice interaction systems. The developed robotic system showed good adaptation in different terrains, and real experiments demonstrated that the robot successfully maintains human social distancing.

A novel social distance mobile approach to automatically detect pairs of humans in a crowded scenario who are not adhering to the social distancing constraint was presented by Sathyamoorthy and colleagues [21]. A mobile robot system is developed, which consists of commodity sensors, an RGB-D camera, and a 2D Lidar to achieve collision-free navigation. Moreover, the developed system involves a thermal camera that transmits thermal images in wireless mode to the healthcare department, so as to monitor the body temperature.

Ramadas and colleagues proposed a social distance monitoring system by employing an automated drone, where the drone transmits alarm signals to nearby police stations and triggers public alarms [22]. The proposed system employs the YOLO-v3 algorithm as an object detection to detect objects efficiently, where YOLO-v3 has the most innovative

version of convolutional neural network of deep networks. In addition, the developed system is able to deliver masks to people who are not wearing them, and gives advice about social distancing and masks.

Standalone monitoring systems are helpful and accurate; however, they pose challenges and have several limitations (coverage and cost). On the other hand, when integrated with the Internet of Things (IoT), wearable devices are expected to offer several advantages, including connectivity, cost, and power consumption.

Table 1 presents a comparison between the existing social distance monitoring systems (wearable units and standalone monitoring units). Wearable units are best for deployability, cost, coverage, and maintenance, whereas the robot-based systems have a high cost and are hard to maintain. On the other hand, fixed camera-based systems are efficient in open environments. However, they offer a low coverage rate, especially in environments that include obstacles and walls.

**Table 1.** A Comparison between the social distance monitoring systems.

| | Wearable Units | Standalone Monitoring Units | |
| | | Fixed Camera | Robots |
| --- | --- | --- | --- |
| Deployment | Easy to deploy, tags are distributed to users | Complicated, requires intensive deployment task, and clear line of sight | Easy to deploy, mobile robots are self-navigated |
| Reliability | High, depends on the employed technology | Medium, due to the presence of walls and obstacles in the area of interest | High, since the robot can navigate the area of interest |
| Coverage | High, depends on the number of available tags | Low depends on the structure of the area of interest, including walls and obstacles | Medium depends on the number of deployed robots |
| Cost | Low, depends on the technology employed in the developed tag | Medium depends on the number of installed monitoring units | High, the robotics technology is still high in cost |
| Maintenance | Easy to maintain, failed tags can be easily replaced | Hard to maintain, monitoring units usually deployed in unreachable positions | Hard to maintain, robot systems are complicated |

## 3. Social Distance Monitoring Approach

This section discusses the design and implementation of a new social distance monitoring system based on social distance tags designed to warn users when approaching people are getting too close in common areas. This section discusses the architecture of the proposed system, including the hardware and software components.

The developed solution is based on two main approaches: human identification and distance measurements. The human identification function is employed to detect the presence of humans approaching the user (who carries the SD-Tag). In contrast, the distance measurement function estimates the distance between the detected people and the user. The overall concept of the proposed social distancing system is shown in Figure 2.

The developed social distance monitoring system consists of four main modules (see Figure 3): human detection, social distance estimation, broadcast, and base-station processing modules.

### 3.1. Human Detection Module

The developed social distance tag (SD-Tag) should be attached to users in public areas (indoor or outdoor) so as to guarantee maintaining social distancing between people in the area of interest. The SD-Tag applies face and eye detection methods to detect the presence of human(s) surrounding the SD-Tag's user. The human detection function is processed by determining the faces or eyes in the surrounding area. This is because most of the women in the Kingdom of Saudi Arabia wear a "Naqab", which shields their faces and, consequently, the face detection method fails to detect the presence of humans. Therefore,

the eye detection method is employed to detect the presence of women who wear a Naqab. Figure 4 shows the concept of the human detection system employed in the proposed social distancing system.

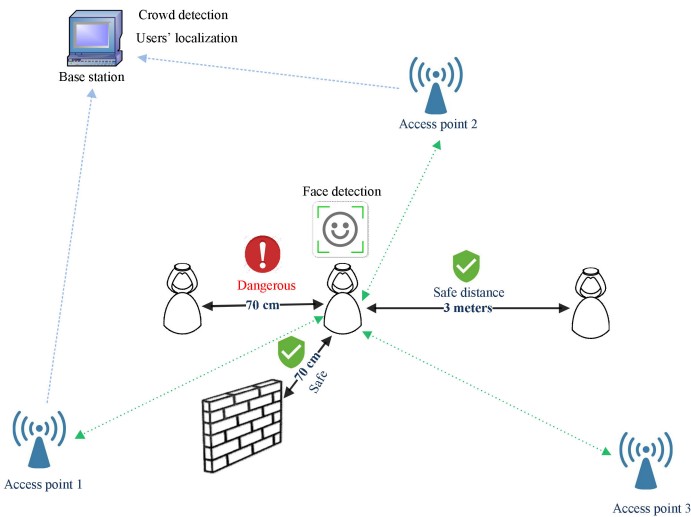

**Figure 2.** The main concept of the proposed social distance solution.

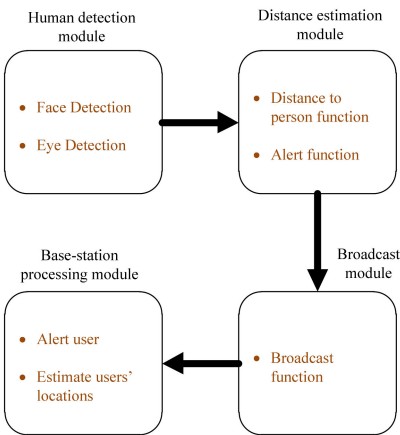

**Figure 3.** Main modules for the proposed social distance monitoring system.

The Haar cascade classifier is implemented for the face detection method, which is an effective method for object detection, as noted by Viola and Jones [23]. Haar cascade is a machine learning-based approach where many positive and negative images are used to train the Haar cascade classifier. In general, the Haar cascade classifier consists of the following:

- Positive Images: These images include the objects the Haar cascade classifier must identify (faces and eyes in our case).
- Negative Images: These images include everything else that does not contain the objects that need to be identified.

On the other hand, the eye aspect ratio (EAR) function has been adopted to compute the ratio of distances between vertical and horizontal eye landmarks [24]. The value from the EAR function will be approximately constant when the eye is open, and decreases towards a zero value during blinking. Algorithm 1 presents the pseudo code for the person detection system.

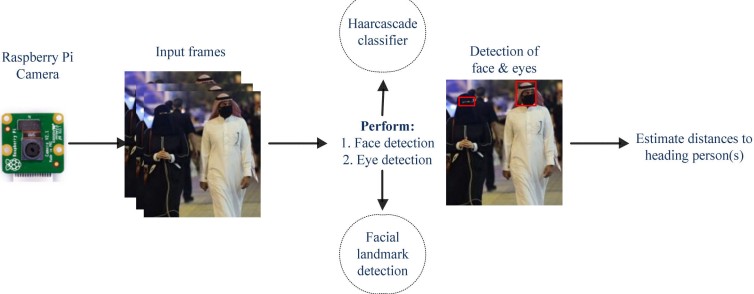

**Figure 4.** The main concept of the human detection system.

---

**Algorithm 1.** Person Detection Algorithm.

---

Input: Array of images from Raspberry Pi camera
Output: Number of persons facing the user
1: let stream is an array of stream bytes received from Pi camera
2: let img is the image formed from the set of *streams*
3: let gray is the gray image
4: let faces is the number of faces captured in a single frame
5: while (stream > 0)
6:     image = camera.capture(stream)
7:     gray = cv2.cvtColor(image, cv2.COLOR_BGR2GRAY)
8:     faces = face_cascade.detectMultiScale(gray, 1.1, 5)
9:     if (faces > 0)
10:      return faces
11:    *else*
12:    *return* null
13: *end*

---

### 3.2. Social Distance Estimation Module

In the second stage, the distance to the detected person is estimated using a range-finder sensor, which can measure the distance between the user (who carries the SD-Tag) and the facing human(s) detected in the first stage. As soon as the SD-Tag obtains a short distance (less than one meter), the SD-Tag will emit warning alerts depending on the distance of the heading person(s) and the number of heading person(s). Algorithm 2 presents the distance estimation algorithm employed in the SD-Tag, and Figure 5 shows the flowchart for the social distance monitoring system.

---

**Algorithm 2.** Distance Estimation Algorithm.

---

Input: Wavelengths emitted by the ultrasonic sensor
Output: Distance values in centimeters
1: let faces is the number of faces received from Algorithm 1
2: let dist is the distance between the user and heading person
3: while (faces > 0)
4:     dist = sensorVal;
5:     if (dist < 100)
6:     alarm_fun(faces, dist)
7:     # the alarm function behaves according to the distance to the
8:     # heading person and the number of heading persons
9: *end*

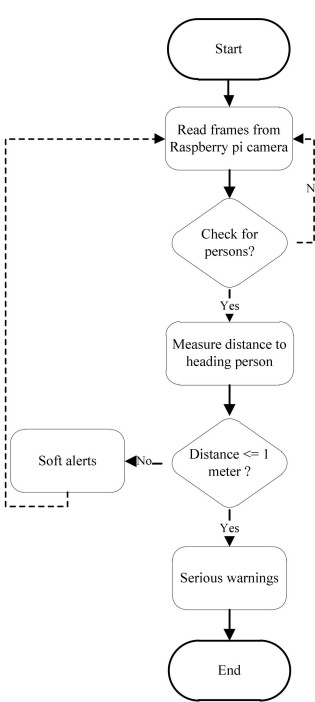

**Figure 5.** The flowchart for the social distance monitoring system.

### 3.3. Localization and Broadcasting Module

The SD-Tag detects the people in the surrounding area and alerts the user (who carries the SD-Tag). In addition, the SD-Tag frequently transmits several parameters to the base station, including the total number of people surrounding the user, estimated distance, current time, location, and access points IDs in the range of the SD-Tag. The localization information is estimated using the system presented by Alhmiedat and Yang [25], and therefore, the SD-Tag can detect the presence of crowds and can alert the base station. Afterwards, users in the same crowded area who carry the SD-Tag will be alerted through continuous beeps. Algorithm 3 shows the pseudo-code for the localization process.

---

**Algorithm 3.** SD-Tag Localization Algorithm.

---

Input: Wifi signals from the surrounding access points
Output: (x, y) coordinates of the SD-Tag$_i$
1: let accessPoint[] is an array of access points that cover SD-Tag$_i$
2: let nAP is the total number of access points (accessPoint.length())
3: let rssAP[] is an array of received signal strength values from accessPoint[]
4: while (nAP >= 0)
5:      rssAP[nAP]=getRSS(accessPoint[nAP])
6:      nAP--;
7: return triangulateLoc(rssAP[])
8: *end*

---

### 3.4. Base-Station Processing Module

Four long-range access points have been employed in different areas to obtain various information from the SD-Tags. The access point transmits this information to the base station, which processes various calculations and obtains notifications from users in the same area. The base station collects the current localization information from each SD-Tag and stores it in an internal database. The base station then checks for crowds (the total number of SD-Tags in a certain sector is more than a predefined threshold), hence warning the SD-Tags' users in that sector. Algorithm 4 shows the processing data algorithm that takes place at the base station.

**Algorithm 4.** Processing Data Algorithm at the Base Station.

Input: location estimation coordinates (x, y) for each SD-Tag
Output: warn all SD-Tags' users who set in a crowded sector
1: let B.S. is the base-station
2: let loc(SD_Tag$_i$) is the 2d location coordinates for SD-Tag$_i$
3: let nSD_Tags is the total number of operating SD-Tags
4: let faces$_i$ is the total number of faces received from SD-Tag$_i$
5: let sector$_j$ is a crowded area in the Park Mall
6: while(nSD_Tags > 0)
7:      SD_Tag$_{nSD\_Tags}$ transmits (x, y) position to BS
8:      if(SD_Tag$_{nSD\_Tags}$ ϵ sector$_i$)
9:      transmit_warnings to SD_Tag$_{nSD\_Tags}$
10:      nSD_Tags--;
11: end

## 4. Experimental Results

This section discusses the experimental testbed and the obtained results. In addition, we discuss the results obtained and compare them with the existing social distance approaches that have been developed recently. Several experiments have been conducted to assess the proposed social distance approach (SD-Tag) by evaluating several parameters.

### 4.1. Experimental Testbed

We tested the proposed system using 33 modules that were developed in the Industrial and Innovation Robotics Center (IIRC) at the University of Tabuk. Figure 6 depicts the architecture for the developed SD-Tag, which consisted of a Raspberry Pi zero w, Raspberry Pi camera, range-finder sensor, accelerometer sensor, and power source (3.6 voltage lithium battery). Table 2 presents the main hardware components employed in our experimental testbed.

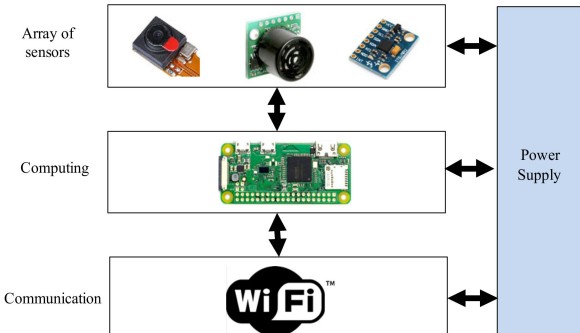

**Figure 6.** SD-Tag architecture.

**Table 2.** The main components of the SD-Tag system.

| Component | Model Name |
| --- | --- |
| Processor | Raspberry pi zero w 1-GHz single core |
| camera | Raspberry pi camera V1.3 |
| Range finder sensor | LV-MaxSonar-EZ0 |
| Power supply | 330 mAH |
| Camera resolution | 280, 160 |

A portable SD-Tag was developed with a size of $7.5 \times 7.5 \times 1.5$ cm$^3$, as presented in Figure 7. The developed system was tested in Tabuk Park Mall, located in Tabuk City, in the Kingdom of Saudi Arabia. Tabuk Park Mall is a quite large mall that consists of almost

280 stores throughout the mall, and it is considered the largest shopping mall in the city of Tabuk. According to the Tabuk Park Mall office, more than 10,000 visitors arrive at the mall daily, and this makes the spread of COVID-19 here likely. Figure 8 shows a side view of the Tabuk Park Mall.

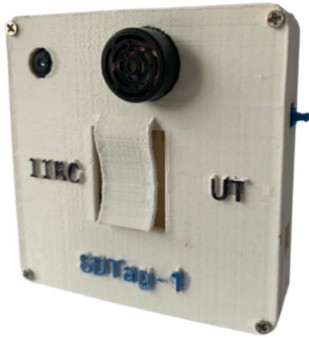

**Figure 7.** The developed SD-Tag.

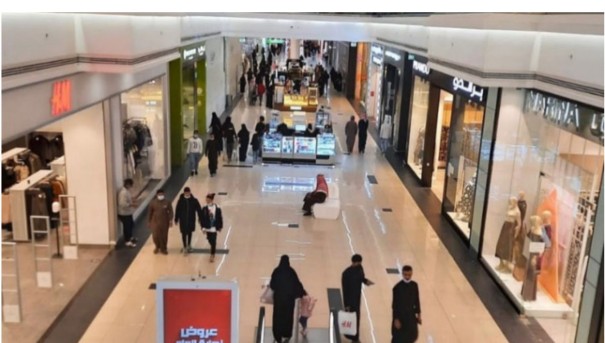

**Figure 8.** Tabuk Park Mall side-view.

The software requirements needed to develop the SD-Tag are as follows: Raspbian as an operating system for the Raspberry Pi zero, OpenCV for processing the images received from the Raspberry Pi camera and then detecting the presence of humans, and Python for developing the social distancing application. Table 3 discusses the characteristics of the vision system employed in the SD-Tag. The SD-Tag device adopts Raspberry Pi camera version 1.3.

**Table 3.** The characteristics of the vision system.

| Specification | Details |
| :---: | :---: |
| Camera name | Raspberry pi camera V1.3 |
| Resolution | 5 megapixels |
| CCD size | $\frac{1}{4}$ inch |
| Field of view | 150 degree |
| Sensor best resolution | 1080p |

### 4.2. Results

A set of 33 volunteers (19 males and 14 females) between the age of 17–52 years old agreed to wear the developed SD-Tag and were then asked to answer a list of questions after completing the shopping task. The experiment took place between 16:00–18:00 p.m. for 3 days, as this period of time is when the park mall is rushed. The performance of the proposed social distance monitoring system was assessed according to the following criteria:

1. User Acceptability: This shows the percentage of users' acceptability for the overall performance of the SD-Tag while traveling inside the Park Mall.
2. User Comfortability: This indicates the user comfortability towards the use of the SD-Tag in the Park Mall.
3. Ease of Use: This shows how easily users can interact with the SD-Tag and be informed through simple warnings and notifications.
4. Social Distance Accuracy: This shows the accuracy of the estimated distance between the SD-Tag user and the heading person(s).
5. Localization Accuracy: This estimates the SD-Tags users' locations, which is a significant issue, in order to position the users located in crowded places.
6. Power Consumption: This estimates the total power consumption for each SD-Tag after accomplishing the shopping tasks in the Park Mall.

SD-Tag users were advised of the importance of these tags for preventing the spread of COVID-19 when the distance between persons is less than 1.5 m. Figure 9 shows the acceptability percentage for the 33 users, with a reasonable average of 96.1%. It shows that the users accepted the developed SD-Tags. The hardware design simplicity led to a wide range of acceptability by several users. In addition, most users accepted wearing such tags as they were well educated about the dangers of getting too close to other people in public spaces, as this increased the chance of spreading the COVID-19 virus among people. In addition, most SD-Tag users advised others to wear SD-Tags during the shopping period.

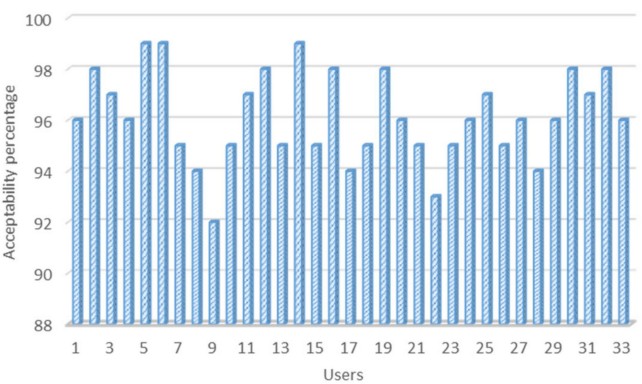

**Figure 9.** Acceptability percentage for 33 users.

Figure 10 presents the ease-of-use percentage for the 33 users with an efficient average rate of 93.3%, which indicates that the developed SD-Tags can be handled easily, with no need for intensive training. The developed SD-Tag consisted of a single push-button which allowed users to switch the device on/off, in addition to several beeps that are emitted from the SD-Tag to warn users of their situations (for instance, continuous beeps means as the user approaches a crowded area). Figure 11 demonstrates the user comfortability of the 33 users when adopting the SD-Tags, where the average user comfortability rate was 92.5%, which indicates that most users were well comfortable when interacting with the SD-Tag, as the developed SD-Tag was small in size and could be easily attached to any user.

As presented in Figure 11, most of the SD-Tags' users were totally interested with the provided functions by the developed social monitoring system. Moreover, most users were satisfied with the warning alerts emitted by the SD-Tag when the SD-Tag user was close to other person(s).

As discussed earlier, social distancing is an important issue in order to maintain safe distance between people in public areas, hence reducing the risk of spreading the COVID-19 virus. Therefore, this section evaluates the social distance accuracy for the SD-Tag device. The social distance accuracy was measured by estimating the difference between the actual distance and the computed distance to the heading person(s). Four different SD-Tag users were involved in this experiment, where the distance between the computed and estimated

distance was measured for each single experiment (eight different experiments for each single SD-Tag user).

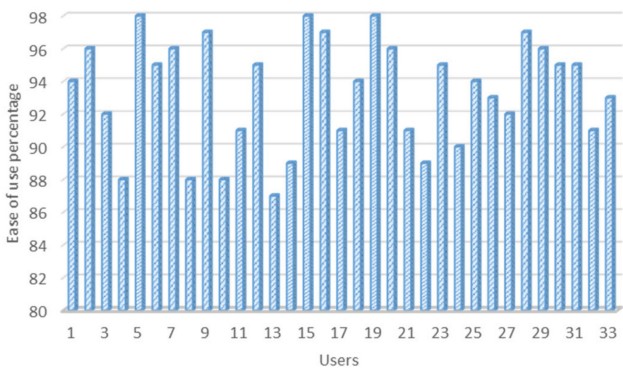

**Figure 10.** Ease of use percentage for 33 users.

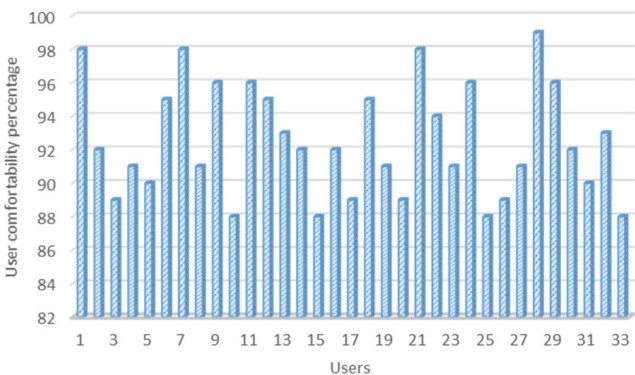

**Figure 11.** Comfortability percentage for 33 users.

As presented in Figure 12, the average of the estimated social distance accuracy for User-A was between 1.18–2.24 m, User-B 1.38–2.12 m, User-C 1.19–2.25 m, and User-D 1.20–1.98 m. Therefore, the average estimated social distance to the heading person(s) using the SD-Tag device was approximately (1.69 m), which indicates that the proposed SD-Tag system offers an efficient estimation measurement to maintain social distancing between people in the area of interest.

On the other hand, the localization accuracy for SD-Tags users has been validated through several experiments, where the SD-Tag has to localize itself every 2 min according to the distributed access points. For many of the experiments, the localization error was measured, which is the difference between the actual position for the SD-Tag user and the estimated position computed by the SD-Tag system. The average localization error was 5.6 m, which was enough to locate crowded areas in the Park Mall, and to then warn the mall security office.

Therefore, the proposed social distance monitoring system consists of two methods to localize the crowded areas: first, calculating the total number of persons recognized by the SD-Tag, and second, by estimating the total number of SD-Tags in a certain area. The integration of the two methods was successful in locating the crowded areas in the Park Mall. Figure 13 shows the average localization error for each SD-Tag user.

The SD-Tag device is equipped with a 3.6-volt lithium rechargeable battery to power the Raspberry Pi zero, camera, and range-finder sensor. The SD-Tag device needs to be recharged regularly (once daily). In this section, we assess the power consumption for the SD-Tag to evaluate the total power consumed through the shopping task. The remaining energy for each SD-Tag is assessed after completing the shopping task by each user. Figure 14 presents the remaining energy in volts for each SD-Tag, with a remaining energy average of 3.01 volt.

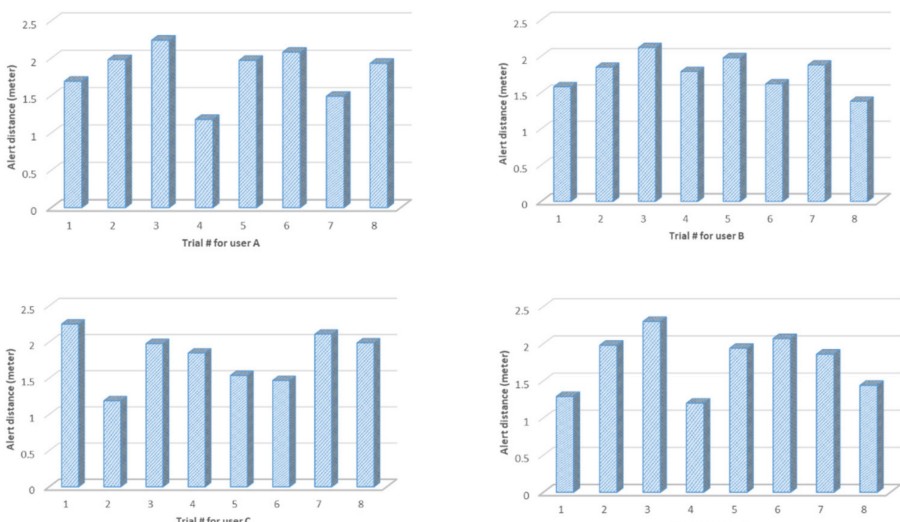

**Figure 12.** Measuring the estimated distance of the SD-Tag to the heading person(s) for four different users.

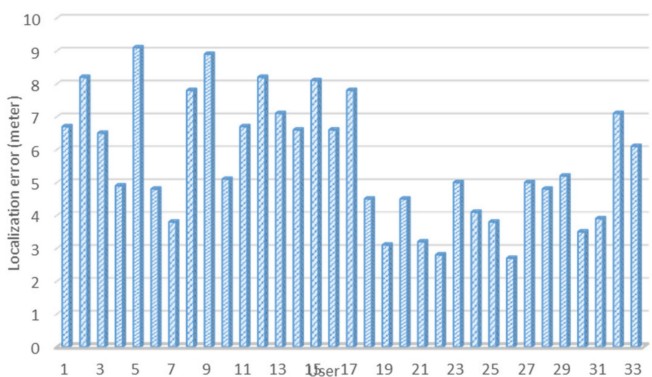

**Figure 13.** The localization error (in meters) for 33 users.

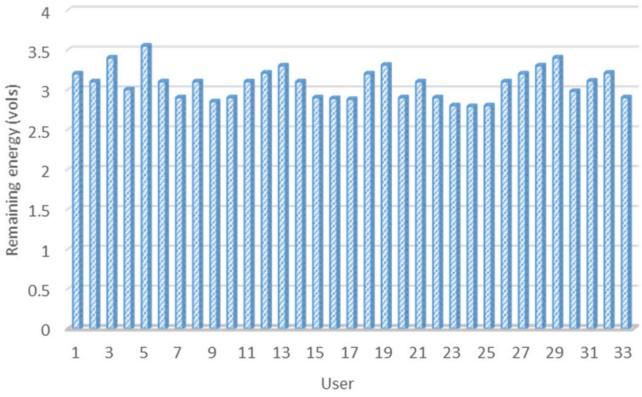

**Figure 14.** The remaining energy (in volts) for each SD-Tag.

*4.3. Discussion*

Maintaining social distancing is the best way to regulate efforts to minimize the spread of COVID-19. Therefore, the developed SD-Tag is an effective social distance monitoring solution that can be employed indoors and outdoors and can minimize the spread of COVID-19 in public spaces. Unlike existing solutions [2–5,7], the proposed system can detect the presence of people in the area of interest and can estimate the distance between the user (who carries SD-Tag) and the surrounding people, with no need to attach an

SD-Tag to every single person in the Park Mall. According to several experiments, only a few users needed to wear the SD-Tag, as the proposed system is based on image analysis, estimating the number of humans in a certain place, and can warn other SD-Tags in the same area.

The deployment of WSN localization systems [11–13] is efficient in cost and power consumption; however, extra cost is added to attach the sensor tags to each user, increasing the overall cost for the developed system. On the other hand, SD-Tag can minimize false alarms in the existing systems proposed in [2–4,8,9]; this is because SD-Tag can recognize the presence of humans in the user's surrounding area.

The works presented in [20–22] are based on robotic systems, which are efficient in terms of maintaining social distances among people in public areas, and are considered attractive solutions. However, these systems are high in cost, and require continuous intensive maintenance. Therefore, in conclusion, the proposed social distance system has the following advantages:

1.  The SD-Tag offers an efficient social distancing method for maintaining social distances between people in public places, with an average accuracy of 1.69 m.
2.  The total cost for the developed SD-Tag is less than 25$, which helps to distribute such tags in different scenarios.
3.  The SD-Tag is a user-friendly device and is easy to interact with, as shown earlier in Figure 10.
4.  The SD-Tag achieves minimum power consumption; the tag can work for the whole day with one charge.
5.  The proposed social distancing monitoring system does not require that all users wear the SD-Tag, as the SD-Tag can monitor the facing area, and then estimate the number of persons in a certain area.

## 5. Conclusions

According to recent statistics, limiting close face-to-face contact with other people is the best way to minimize the spread of COVID-19 disease. In this paper, we developed a new social distance system that limits the spread of COVID-19 in crowded places. The main contribution of the proposed work is that it efficiently and cost-effectively guarantees safe social distancing between people indoors. The developed SD-Tag has been validated through several experiments, and achieved reasonable accuracy and user acceptability. We aim to employ wireless sensor networks to guarantee high localization accuracy, minimum cost, and low power consumption for future works.

**Author Contributions:** M.A. proposed the main concept of the social distance monitoring system; in addition, he analyzed and discussed the recent developed research works focused on the social distance monitoring systems. T.A. built the social distancing monitoring system and performed the implementation task. Moreover, T.A. accomplished several real experiments to validate the performance of the proposed social monitoring system, and analyzed and discussed the obtained results. All authors have read and agreed to the published version of the manuscript.

**Funding:** This research was funded by the Deanship of Scientific Research, University of Tabuk, Tabuk, Saudi Arabia, under grant number S-501-1440.

**Acknowledgments:** The authors would like to acknowledge the financial support for this work received from the Deanship of Scientific Research, University of Tabuk, Tabuk, Saudi Arabia, under grant number S-501-1440.

**Conflicts of Interest:** The authors declare no conflict of interest.

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
