# Peer review of "Social Distance Monitoring Approach Using Wearable Smart Tags"

_electronics, doi:10.3390/electronics10192435_

Round 1
Reviewer 1 Report
Dear author,
the paper sounds interesting for the community. However, you should add more information about the scientific literature about this technology (not only about yours) and about COVID using scientific references. Then, I suggest recreating a structure between methods, discussion and results. In addition, when you speak about population then you should also give information about the anthropometry (gender, age, ....)of this population.
Finally, you should also study statistically more your data to prove that you device is important or accepted/or not by your population.
All this will help readers to understand the importance of this research.
Author Response
The corrections are made in Green color font.
Please see the attachment.

Reviewer 2 Report
This manuscript proposes an approach with wearable tag to monitor the social distance in the park mall to prevent the spread of COVID-19. The manuscript provide the comprehensive literature review and a well-organized description of the proposed approach and developed system. Experiments are conducted and the results validate that the proposed approach achieves the satisfactory performance, including user acceptability, user comfortability, ease of use, etc.
The main concern of the reviewer is the localization accuracy. According to the experimental result, the propose system generates approximately 5.6 meters localization error in average. However, it presents a disappointing result because the proper social distancing measures for COVID-19 are 1.5 and 1 meters in indoor and outdoor environments, respectively.
Author Response

(The authors gave the same response as above.)

Round 2
Reviewer 1 Report
Dear author,
I just to publish it after your revision.